# Current Hydration Habits: The Disregarded Factor for the Development of Renal and Cardiometabolic Diseases

**DOI:** 10.3390/nu14102070

**Published:** 2022-05-15

**Authors:** Richard J. Johnson, Fernando E. García-Arroyo, Guillermo Gonzaga-Sánchez, Kevin A. Vélez-Orozco, Yamnia Quetzal Álvarez-Álvarez, Omar Emiliano Aparicio-Trejo, Edilia Tapia, Horacio Osorio-Alonso, Ana Andrés-Hernando, Takahiko Nakagawa, Masanari Kuwabara, Mehmet Kanbay, Miguel A. Lanaspa, Laura Gabriela Sánchez-Lozada

**Affiliations:** 1Division of Renal Diseases and Hypertension, University of Colorado Anschutz Medical Campus, Aurora, CO 80045, USA; richard.johnson@cuanschutz.edu; 2Department Cardio-Renal Physiopathology, INC Ignacio Chávez, Mexico City 14080, Mexico; enrique.garcia@cardiologia.org.mx (F.E.G.-A.); ggonzaga49@gmail.com (G.G.-S.); alejandro.vlz.orozco@gmail.com (K.A.V.-O.); yamniaalvarezalvarez@gmail.com (Y.Q.Á.-Á.); emilianoaparicio91@gmail.com (O.E.A.-T.); ediliatapia@hotmail.com (E.T.); horace_33@yahoo.com.mx (H.O.-A.); 3Division of Nephrology and Hypertension, Oregon Health Sciences University, Portland, OR 97239, USA; andreshe@ohsu.edu (A.A.-H.); lanaspa@ohsu.edu (M.A.L.); 4Department of Nephrology, Rakuwakai Otowa Hospital, Kyoto 607-8062, Japan; nakagawt@gmail.com; 5Intensive Care Unit, Toranomon Hospital, Tokyo 105-8470, Japan; kuwamasa728@gmail.com; 6Department of Cardiology, Toranomon Hospital, Tokyo 105-8470, Japan; 7Division of Nephrology, Department of Internal Medicine, Koc University School of Medicine, Istanbul 34010, Turkey; mkanbay@ku.edu.tr

**Keywords:** underhydration, water intake, metabolic syndrome, obesity, chronic kidney disease

## Abstract

Improper hydration habits are commonly disregarded as a risk factor for the development of chronic diseases. Consuming an intake of water below recommendations (underhydration) in addition to the substitution of sugar-sweetened beverages (SSB) for water are habits deeply ingrained in several countries. This behavior is due to voluntary and involuntary dehydration; and because young children are exposed to SSB, the preference for a sweet taste is profoundly implanted in the brain. Underhydration and SSB intake lead to mild hyperosmolarity, which stimulates biologic processes, such as the stimulation of vasopressin and the polyol-fructose pathway, which restore osmolarity to normal but at the expense of the continued activation of these biological systems. Unfortunately, chronic activation of the vasopressin and polyol-fructose pathways has been shown to mediate many diseases, such as obesity, diabetes, metabolic syndrome, chronic kidney disease, and cardiovascular disease. It is therefore urgent that we encourage educational and promotional campaigns that promote the evaluation of personal hydration status, a greater intake of potable water, and a reduction or complete halting of the drinking of SSB.

## 1. Introduction

Healthy lifestyle factors provide benefit by both preventing and slowing the progression of non-communicable diseases [1]. While the role of a healthy diet and exercise are well-recognized [1], the role of water intake has been largely disregarded [2]. However, the intake of drinking water is critical to the maintaining of the osmolarity and concentrations of electrolytes needed to allow critical biochemical reactions to take place in both the intracellular and extracellular environment [3]. While it is well-recognized that the host can reduce water losses and accommodate in the setting of mild dehydration to maintain serum osmolarity, only recently has it been recognized that the chronic stimulation of biological responses to maintain osmolarity can have deleterious consequences on health [4,5]. This suggests that ‘underhydration’, in which the activation of these biological processes results in normal serum osmolarity, may represent an unrecognized but significant risk factor for disease. Until now, there has been some debate on this topic, as studies have found inconsistent findings [6]. Here, we briefly review the evidence presented by randomized trials, epidemiological studies, and basic science, indicating that suboptimal hydration, by chronically triggering the mechanisms aimed to preserve body water and protect against hypertonicity, i.e., vasopressin and the aldose reductase pathways, may be partially responsible for an increased tendency to develop chronic non-communicable diseases at younger ages.

## 2. Types of Dehydration

Dehydration usually refers to a loss of water resulting in an increase in serum and intracellular osmolarity (hypertonic dehydration [Na^+^]≥ 150 mEq/L or ≥ 300 mOsm/L), and it is associated with a reduction in intracellular volume [7]. Most commonly, dehydration occurs from either fluid restriction or water loss (such as in sweat or diarrhea) [7].

A biologically equivalent dehydration state may also occur through two other mechanisms. The first is by the ingestion of salt in the absence of drinking sufficient water to maintain isotonicity, leading to a condition of hyperosmolarity and thirst [8]. The other mechanism is one that leads to a relative shift of water from the extracellular to the intracellular environment. One means is by drinking fructose-rich beverages, which leads to hyperosmolarity by shifting water into the intracellular space, likely due to the increased intracellular glycogen synthesis that adsorbs water [9].

While dehydration refers to a relative reduction in the water to salt concentration in the extracellular environment, one can also lose salt, which is important in maintaining extracellular volume. When the extracellular volume level is low, it can lead to extracellular volume depletion and hypotension. Extracellular volume (ECV) depletion may also result in a greater loss of water, leading to both an increased serum osmolarity and volume depletion (hypertonic volume depletion with serum sodium ≥145 mEq/L), and then to volume depletion with normal serum osmolarity (isotonic volume depletion with serum sodium [135–144 mEq/L.]) or with a relatively greater loss of salt to water (hypotonic volume depletion with serum sodium <135 mEq/L). While dehydration is well-recognized in medicine, recently the condition of “underhydration” was coined by Kavouras [10] and further defined by Stookey as having serum sodium >145 mmol/L, a first void urine volume <50 mL, and/or spot urine osmolality ≥500 mOsm/kg [11]. In this condition, serum osmolality is maintained at the upper end of the normal range [270–290 mOsm/kg water] at the cost of the chronic elevation of vasopressin and the activation of the renal mechanisms aimed toward the conservation of body water.

Underhydration is a common condition. A recent study found that over 95% of US adults (51–70 years old) were underhydrated; this condition was cross-sectionally associated with the increased prevalence in obesity, metabolic syndrome, insulin resistance, diabetes, high waist circumference, hypertension, and low high-density lipoproteins. After three to six years of follow-up, underhydrated subjects had a 4.21-fold higher risk of dying of chronic diseases [12]. Moreover, an analysis of a NHANES III sample reported a prevalence in elevated serum osmolality of 60% in older adults (≥295 mmol/L) [13]. 

## 3. What Constitutes an Adequate Water Intake?

In resting humans, there is a continuous loss of water through the process of respiration and through the skin; there is also an intermittent loss of water and electrolytes through the urinary and gastrointestinal tracts. Such a loss of water and electrolytes can be amplified by exercise and heat stress, as well as by not replacing such water losses. The oral intake of liquids and food is the primary means in humans by which to replenish water losses. Despite the importance of an adequate hydration status, there is not a worldwide consensus regarding how much water people should drink daily to maintain a healthy state of hydration [14]. 

Recently, recommendations for an adequate intake of water have been provided by the European Food Safety Authority (EFSA) [15] and by the Institute of Medicine (IOM) [16] in the USA (Table 1). In most countries, up to 80% of total water intake comes from fluid intake; therefore, the drinking of beverages represents the primary means to modulate total water intake.

Based upon these recommendations, several epidemiological studies have found that the intake of water in many populations is inadequate. The NHANES III survey data, for example, estimated the prevalence of dehydration in adults to be 16–28% [13]. Another study that examined data from 13 countries from three continents suggested that underhydration was even more prevalent, with only 40% of males and 60% of females having a water intake within the recommended guidelines of the EFSA [17]. Another large cross-sectional survey that included 15 countries from three continents found that 47% of adults, 60% of children, and 75% of adolescents do not meet the EFSA recommendations for adequate fluid intake [18]. In addition, the use of biomarkers to classify hydration status suggests that water intake may be inadequate [19,20]. Subjects with obesity are also at a higher risk for being dehydrated than lean subjects [19]. 

Nevertheless, all of these studies carried some bias as they were based on self-completed surveys. On the other hand, the analysis of fluid intake based upon the biological markers of hydration status—such as serum or urine osmolality; plasma sodium concentration; urine volume, color, and specific gravity; and body weight changes—can provide more reliable data about body fluid composition. Based on such types of analyses, normal hydration has been defined as a urine osmolality between 500 mOsm/kg H_2_O and 800 mOsm/kg H_2_O [21,22,23]. Urine-specific gravity is another surrogate for hydration status that correlates with urine osmolality; a urine-specific gravity of 1.020 is considered the upper limit for normal hydration [24,25]. An easier way to detect hydration status that has gained popularity is the urine color chart developed by Armstrong for adults and Kavouras for children [26,27]. This method is an inexpensive and practical tool for evaluating hydration status and has been validated to correlate with body water deficits and to reflect changes in body water.

Low water consumption may be voluntary or involuntary [28,29]. Voluntary dehydration occurs despite adequate water availability and may relate to work conditions, lack of public toilet facilities, an inadequate response to thirst, lack of motivation, lack of a preference for water, and/or a preference for a sweet taste [30,31,32,33,34]. Voluntary dehydration in young children represents a special case, as they have a higher proportion of body water, making up 75% of their total body weight [35]; they also depend upon parents’ or caregivers’ encouragement to drink water. Thus, this population is especially at risk for underhydration [33]. On the other hand, involuntary dehydration is associated with the unavailability of clean tap water for intake [36] or an increased threshold for thirst, as happens with aging [37].

## 4. Intake of Sweetened Beverages

The intake of fluids is highly affected by flavor both in children and adults [38,39,40]. Thus, children and adolescents have a marked preference for a sweet taste that is strengthened by early and frequent intake [41]. It has been shown that the consumption of sugar- sweetened beverages (SSB) during infancy is a predictor of later intake of SSB [42]. Experimental studies have shown that SSB intake during early life makes animals such as rats more susceptible to severe kidney injury following a bout of acute kidney injury in comparison to rats that received water [43]. Moreover, it has been reported that the stronger the preference for a sweet taste, the higher the intake of SSB [34]. 

The preference for sweets coupled with frequent early exposure to SSBs likely contribute to the increase in SSB consumption worldwide [44]. SSB include carbonated soft drinks, fruit drinks/juice, sports/energy drinks, and sugared coffee/tea drinks. In many regions, there are also culturally specific SSB, as is the case for “aguas frescas” in Mexico and some other Latin American countries [45]. Globally, North America and Latin America are the largest consumers of SSB [46]. SSB intake especially affects poorer communities, where it accounts for as much as from 10% to 23% of total calorie consumption [47].

In the United States, the intake of SSB has been decreasing in recent years [48]. However, more detailed analysis has shown that there are both regional and sociodemographic differences in SSB consumption patterns. High consumption rates for SSB remain in the South [49] and are also found among specific groups, including young adults, non-Hispanic blacks, Mexican Americans, and children and adults from low-income families [50,51]. In Latin America, more than 80% of Brazilian and Mexican adults drink one or more SSB per day, with lesser rates reported in Argentina and Uruguay [52]. On the other hand, Chile has the fastest absolute growth in SSB sales throughout the world [46].

High-income and middle-income countries have developed strategies to curtail the intake of SSB, including taxation, banning the sale of SSB in schools and other public institutions, restrictions on the marketing of SSB to children, front package labels, and public awareness campaigns [46]. Notably, a tax between 10 and20% has been shown to be successful in decreasing the purchasing of SSB, while a tax of 30% markedly discourages the purchasing of SSB [53]. In Mexico, an SSB taxation rate of 10% resulted in a decrease of 9.5% in sales after two years, with the largest decrease found amongst the most socially disadvantaged households [54]. Likewise, the 25% tax imposed on SSB in Berkeley, California resulted in a 21% fall in sales in low-income neighborhoods within the first four months [55]. However, long-term epidemiological studies are needed to firmly establish whether taxation has the potential to reduce SSB intake. An additional consequence of taxing SSB is the encouragement to reformulate these products using a combination of sugars and nonnutritive sweeteners (NNS) to maintain the sweet flavor. Such action has unknown consequences in the long term, as there is some evidence that NNS can alter gut microbiota, brain response, and heart health [56].

High SSB intake is the main contributor to added sugar in the diet. Currently, there is strong evidence relating SSB intake to obesity, with an additional increased risk for those who have a genetic predisposition. Cardiometabolic diseases, metabolic syndrome, type 2 diabetes, non-alcoholic fatty liver disease, hypertension, gout, some forms of cancer, and cardiovascular mortality are also positively associated with SSB intake [57]. On the other hand, a well-designed clinical trial found that replacing SSB with non-caloric beverages diminished weight gain and fat accumulation [58]. However, it is important to notice that the deleterious effects of SSB are also causally linked to metabolic effects, such as fatty liver and diabetes, independently of weight gain.

## 5. Metabolic, Cardiovascular, and Renal Effects Resulting from Poor Hydration Habits

Underhydration due to low water intake or recurrent mild dehydration is associated with the chronic activation of the renin angiotensin system, endothelin, vasopressin, and the aldose reductase-fructokinase (AR-F) pathway [59,60]. While the renin-angiotensin and endothelin systems have been better characterized in acute hypohydration [61], the present review will focus on the role of the vasopressin and AR-F pathways, as well as their interactions and synergistic effects.

### 5.1. Vasopressin Pathway

Vasopressin is a peptide hormone released by the posterior pituitary in response to an increase in plasma osmolality or a decrease in blood pressure. The primary known purpose of vasopressin is to decrease water loss by concentrating the urine through a mechanism that involves binding the V2 receptors in the collecting duct of the kidney [62]. There are also V2 receptors in the lungs, where they might reduce the loss of water vapor [63]. Thus, vasopressin is a critical hormone involved in protecting the organism from dehydration.

A recently recognized, novel role for vasopressin in the conservation of water has been that vasopressin may mediate the synthesis and storage of fat as a means of providing metabolic water [64]. Specifically, many animals, such as whales and desert mammals, metabolize their body fat as an additional source of water when fresh water is not easily available [65].

In humans, copeptin (the stable vasopressin analogue) concentrations in plasma have been found to be significantly associated with increased risk of type two diabetes, chronic kidney disease, and cardiovascular disease [66,67,68,69]. Moreover, low water intake, which induces chronic vasopressin secretion [70], predicts the development of risk factors for coronary artery disease, for example, hyperglycemia [71]. On the contrary, increasing water intake, a maneuver that decreases systemic copeptin concentrations [72,73], was found to protect against the development of metabolic syndrome and fatty liver in fructose-fed mice and in obese Zucker rats [64,73]. In humans, increasing water intake has also been associated with reduced concentrations of copeptin, fasting glucose, glycated Hb in men, and a lower type 2 diabetes risk [72,74,75,76]. Increased water intake has also been reported to lead to weight loss [77].

Chronic high levels of vasopressin may also have a role in driving kidney damage. There is early evidence that the chronic stimulation of vasopressin can accelerate kidney disease [78]. Experimentally, the chronic administration of desmopressin (a vasopressin V2 receptor agonist) induced albuminuria and kidney damage in rodents and humans [79,80]. Similarly, stimulation of the V1a receptor may cause renal vasoconstriction [81] as well as gluconeogenesis and glycogenolysis [5]; and coronary artery vasoconstriction [81].

Increased water intake, on the other hand, reduces the risk of albuminuria and CKD [82,83]; is inversely associated with CKD prevalence [84]; and is also associated with a favorable lipid profile, independently of physical activity and the intake of fats, protein, or fiber [85]. Despite these findings, in a randomized trial, stage three CKD patients reported that increasing water intake did not slow the progression of the disease. However, the increase in water intake achieved was relatively small (700 mL), and the study was underpowered to detect differences based on the small changes in copeptin (vasopressin) levels observed (−7% in coached patients) [86]. 

### 5.2. Aldose Reductase-Fructokinase (AR-F) Pathway

This pathway is mainly activated by hypertonic stress, hypoxia, and hyperglycemia [87]. Other mechanisms include hyperuricemia [88,89] and underhydration [43]. Overexpression of AR diverts glucose towards the aldose reductase pathway with the generation of sorbitol and fructose catalyzed by sorbitol dehydrogenase. The increase in endogenous fructose induces the activation of fructokinase, which is the limiting enzyme in fructose metabolism [90]. Increased fructokinase activity induces a temporary depletion in ATP from the synthesis of fructose-1P, because fructokinase metabolizes fructose so rapidly, in contrast to hexokinase [91]. Such an effect also depletes intracellular phosphorous, which induces the activation of AMP-deaminase 2 and the downstream degradation of purines with the increased synthesis of uric acid. This in turn inhibits AMP-kinase, inhibits aconitase-2 and enoyl-CoA hydratase in the Krebs cycle, activates NADPH oxidase, and diminishes nitric oxide bioavailability [92,93,94]. In the liver, these effects favor de novo lipogenesis and block fatty acid oxidation, increase oxidative stress, promote insulin resistance, and induce liver steatosis and metabolic syndrome [92]. Increased serum osmolality induced by a high-sodium diet also activates the AR-F pathway and results in obesity, metabolic syndrome, and fatty liver [8].

In the kidney, dehydration may also activate the polyol-fructose pathway, where it may mediate kidney injury. Specifically, while aldose reductase is constitutively expressed in the renal medulla—where it produces sorbitol that acts as an osmotic agent to protect tubular cells from the hyperosmolar environment—it is normally not expressed in the renal cortex [87]. However, with dehydration, this enzyme is induced, and along with sorbitol dehydrogenase, results in fructose generation [87]. The proximal tubule contains fructokinase and can metabolize the fructose, leading to local oxidative stress and the release of inflammatory chemokines [95]. Local tubular damage can result and, in the setting of chronic heat stress and dehydration, may lead to chronic kidney disease that is fructokinase-dependent [96]. It has been proposed that this might be a mechanism for explaining the epidemics of CKD of unknown etiology that have emerged in Central America and elsewhere [97]. Thus, the activation of the AR-F pathway is also partially responsible for diabetic and recurrent dehydration nephropathies as well as acute kidney injury [43,90,96,98]. In addition, the uric acid generated during fructose metabolism induces endothelial dysfunction by decreasing the nitric oxide bioavailability and induces endothelial cell senescence while promoting vascular muscle cell hypertrophy, renin-angiotensin activation, and the systemic elevation of protein C reactive; such effects likely contribute to the development of hypertension [94,99]. 

In the heart, an increased AR-F pathway may also occur in response to ischemia from causes such as coronary artery disease, and there is evidence that this is responsible for cardiac remodeling and eventually, hypertrophy [100,101,102]. Collectively, the deleterious effects induced by the chronic activation of the AR-F pathway increase the probability of developing cardiovascular, renal, and metabolic diseases.

### 5.3. Synergy of the Vasopressin and AR-F Pathways

Both vasopressin and the AR-F pathways act in synergy, aggravating target organ damage. In fact, fructose has been found to directly stimulate vasopressin secretion [103,104]. SSB intake increases vasopressin (copeptin) concentrations in rats and humans exposed to heat stress, and it is associated with markers of kidney damage [60,105]. Wolf showed that a fructose infusion given to healthy men augmented vasopressin secretion, an effect not induced by an isotonic solution of glucose. Thus, the mechanism by which fructose stimulates vasopressin does not appear to become from its effects on osmolarity [103], but rather from the metabolism of fructose in the hypothalamus [104]. Specifically, acutely dehydrating mice has been shown to induce the expression of aldose reductase of the polyol pathway in the supraoptic nuclei of the hypothalamus, followed by the production of fructose. In turn, the fructose is metabolized locally by fructokinase, which drives the synthesis of vasopressin [104]. Thus, mice lacking fructokinase show an impaired vasopressin response to dehydration [104]. Moreover, the oral ingestion of fructose in both laboratory animals [60,64,106] and humans [105,107] results in a rise in vasopressin (copeptin) levels in the blood. Thus, fructose metabolism is directly linked with the stimulation of the vasopressin system (Figure 1). 

Consistent with these findings, recurrent heat-induced dehydration induced a lower secretion of copeptin in fructokinase knockout mice (FKM) compared to wild type mice, despite both groups having similar serum osmolality [96]. FKM were also protected from developing dehydration-induced chronic kidney disease. Importantly, it has been shown that, in both dehydrated and non-dehydrated rodents, plasma copeptin dose-dependently increased with fructose concentrations in drinking fluid without modifying serum or urine osmolality [64,108]. The stepped dose–response of copeptin to fructose concentrations was also associated with a stepped decline in kidney function [108] and stepped body adiposity [64]. 

We also found that fructose-induced metabolic syndrome was mediated by elevated vasopressin levels, which acted by stimulating the V1b receptor [64]. The mechanism is not totally known, but it may involve the stimulation of glucagon (via V1b receptors on the pancreatic islets) by the stimulation of ACTH from the V1b receptors on the anterior pituitary, or perhaps by direct stimulation of the liver itself [64]. Indeed, the stimulation of vasopressin 1b receptors was shown not only to mediate fat production (as a source of metabolic water) but also other features of the metabolic syndrome, thus suggesting a potential role for underhydration in the development of obesity and diabetes [64]. The relevance of this finding is that people who are obese are frequently hyperosmolar and have high vasopressin levels (noted by elevations of copeptin, the stable analogue), likely because of diets rich in sugar and salt [109,110]. Underhydration is also substantially more common in this group than among the general population [12,19]. 

In this regard, we have found that a high salt diet (or dehydration itself) can induce both fructose and vasopressin formation [64,104,107]. Moreover, high salt intake may lead to increases in serum osmolality, thereby inducing chronic underhydration, activating both the polyol-fructose and vasopressin pathways. While high salt intake is linked with hypertension and cardiovascular events, salty diets also increase the risk for obesity and metabolic syndrome [8,110]. When laboratory animals are placed on a salty diet, they initially appear to be catabolic, but after a few months, they dramatically gain weight and develop the features of metabolic syndrome. The mechanism was shown to be mediated by the induction of the polyol-fructose pathway [8], and likely the vasopressin system as well. Consistent with these findings, we were able to show that a high level of salt intake also correlated with an increased risk for hepatic steatosis and for diabetes [8].

The AR-F and vasopressin pathways may also be involved in the pathogenesis of primary hypertension. As these pathways are driven more by osmolarity than by salt content per se, it is of interest that serum sodium concentration is a better predictor for hypertension than total sodium intake [111]. Likewise, the acute ingestion of salt raises blood pressure and vasopressin levels in humans, but if it is accompanied by water intake that suppresses the rise in serum osmolarity, no rise in blood pressure or vasopressin is observed [112]. Likewise, both fructose and uric acid (its metabolite) have been found to predict the development of hypertension [113,114]. Additionally, lowering uric acid has been reported to block the hypertensive response to fructose [115] and also to correct blood pressure in adolescents presenting with newly diagnosed primary hypertension [116,117]. 

Low water intake and increased consumption of SSB are habits which are typically rooted in infancy. We reported that underhydration induced by the chronic restriction of water intake in infant rats induced a greater secretion of copeptin and increased fructokinase expression in the kidney [43]; this effect was further amplified when rats were hydrated with SSB after water restriction. Moreover, in those rats, an event of acute kidney injury induced a catastrophic reduction of creatinine clearance and mortality, while animals rehydrated with water better endured the insult [43]. These data suggest that poor hydration habits activate vasopressin and AR-F pathways and that synergistically, these pathways lead to worse renal outcomes after an acute insult.

There is also evidence that vasopressin activates the AR-F pathway. For example, a chronic vasopressin blockade with conivaptan or tolvaptan prevented the overexpression of the components of the AR-F pathway in recurrently mildly heat-dehydrated rats that were rehydrated with 10% fructose and protected them from kidney damage [106,118]. Thus, vasopressin also regulates the expression of AR-F enzymes, at least in the context of mild dehydration. The exact mechanism concerning how vasopressin stimulates the AR-F pathway is not yet known, but the V2 receptor stimulation of cAMP and protein kinase A (PKA) can activate the tonicity-responsive enhancer/osmotic response element-binding protein (TonEBP/NFAT5) which regulates AR [119]. On the other hand, supplementation with antioxidants prevented the upregulation of AR-F enzymes in mildly dehydrated rats that were rehydrated with fructose [120] by preventing the nuclear translocation of the transcription factors responsible for AR and F, which are NFAT5 and ChREBP, respectively [120]. Since tolvaptan activates the nuclear factor erythroid 2-related factor (Nrf2) antioxidant pathway through the phosphorylation of protein kinase RNA-like endoplasmic reticulum kinase (PERK) [121], this can explain why this treatment also prevented the up-regulation of AR-F enzymes in a state of mild dehydration [118]. Such data strongly suggest that increased oxidative stress plays an important role in the crosstalk between the vasopressin and AR-F pathways. A summary of the interactions between vasopressin and AR-F pathways is depicted in Figure 1.

## 6. Conclusions and Recommendations

Physicians, and especially nephrologists, are trained to recognize dehydration and volume depletion and their acute health consequences. However, only recently has the concept of underhydration been recognized. For years, many of us were taught that the biological responses to dehydration that led to a correction of the dehydrated state simply reflected the exquisite ability of the kidney to ensure homeostasis, and the adage to drink a lot of water every day was not necessary to follow when such an eloquent organ as the kidney could fine tune our metabolism with ease. However, what we have learned is that the corrective pathways to maintain serum osmolarity, when they are chronically activated, carry a consequence, and this translates into an increased risk for obesity, metabolic syndrome, hypertension, and chronic kidney and heart disease. So, the concept of “hydration for health” should take on new meaning for the medical community today.

Therefore, we recommend educational campaigns to encourage people to evaluate their personal hydration status, for example, by checking urine color, and to encourage a greater intake of potable water. We also suggest educating parents and caregivers to prevent offering SSB during infancy and young childhood, as at these young ages the preference for a sweet taste is deeply implanted in the brain.

## Figures and Tables

**Figure 1 nutrients-14-02070-f001:**
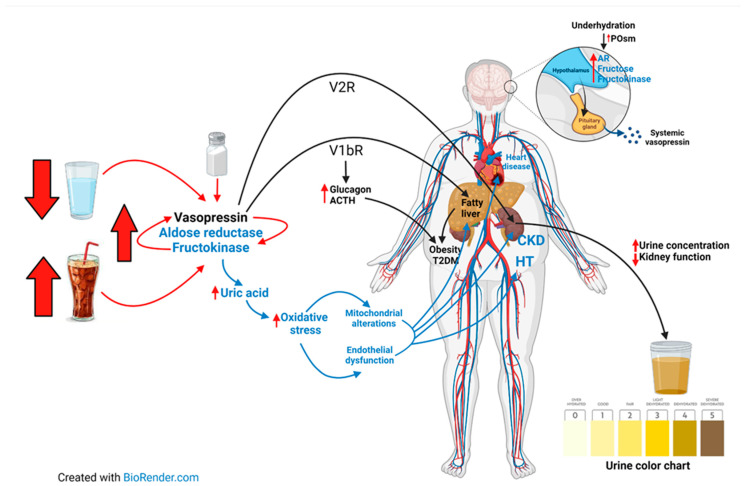
Low water intake coupled with sugar-sweetened beverage (SSB) intake activate both vasopressin and the aldose reductase-fructokinase pathways, which also can synergize with each other, aggravating target organ damage. In the brain, milder increases in systemic osmolality due to dehydration induce the expression of the aldose reductase pathway in the supraoptic nuclei of the hypothalamus, followed by the production of fructose. In turn, the fructose is metabolized locally by fructokinase, driving the synthesis of vasopressin. The increased vasopressin activity stimulates V2R in the kidney, inducing urine concentration and chronically renal alterations. Stimulation of the V1b receptor occurs in the pancreatic islets as well as in the liver and the anterior pituitary, thereby stimulating the secretion of glucagon and the adrenocorticotropic hormone. V1bR stimulation also mediates fat production and other features of metabolic syndrome, such as fatty liver, obesity, and eventually, type 2 diabetes. Aldose reductase-fructokinase pathway activation induces an increase in uric acid synthesis which induces intracellular oxidative stress, mitochondrial alterations, and endothelial dysfunction. This eventually results in target organ damage, the organs that are more affected being the liver (fatty liver), heart (heart failure), kidney (chronic kidney disease) and the vessels (hypertension). High salt intake may lead to an increase in serum osmolality, thereby inducing chronic underhydration, activating both the polyol-fructose and vasopressin pathways. (Abbreviations: ACTH, adrenocorticotropic hormone; AR, aldose reductase; CKD, chronic kidney disease; HT, hypertension; POsm, Plasma osmolality; T2DM, type 2 diabetes mellitus).

**Table 1 nutrients-14-02070-t001:** Adequate intake for fluids (mL/day) accordingly to EFSA [9] and IOM [10].

Age	EFSA	IOM
	mL/day	mL/day
0–12 months	680–800	700–800
1–3 years	1000	900
4–8 years	1200	1200
9–13 years:		
Male	1600	1800
Female	1500	1600
>14 years and adults:		
Male	2000	2600
Female	1600	1800

## Data Availability

Not applicable.

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
