# Peer review of "Current Hydration Habits: The Disregarded Factor for the Development of Renal and Cardiometabolic Diseases"

_nutrients, 2022, doi:10.3390/nu14102070_

Round 1

Reviewer 1 Report

This paper contributes to the hydration literature by helping to resolve controversy stemming from unchecked assumptions about homeostatic compensation for hyperosmolality. By succinctly summarizing data that show that the compensations are not benign, the paper warrants greater attention to indices of hyperosmotic stress and intervention potential to reduce chronic disease risk. Noting that the harms or costs related to compensation for hyperosmolality are preventable.

A few edits might clarify the purpose and strengthen the contribution of this paper. The controversy and its implications (e.g. inconsistent findings, health professional inaction, missed intervention opportunities) might be introduced in the background section. The methods for or logic behind highlighting data points that demonstrate a harmful effect of compensation might be made more transparent. The introduction might outline or foreshadow what kinds of data will be presented or what data questions were considered to sway the reader’s opinion… Randomized trials, epidemiological data, plausible causal paths confirmed with knock out models.

Low water or high salt à hyperosmolality-> vasopressin ->receptors-> pathways -> chronic disease

Do we have evidence of each link in this causal chain?  Is each of these steps in the mechanism prevalent in free-living populations?  If these are/were the authors implicit working questions, the paper might make that explicit. Explicit questions up front might give more context to how the headers fit together into the whole.  (Types of dehydration, what constitutes adequate water intake, causes of inadequate water consumption, intake of sweetened beverages, metabolic…effects of poor hydration)

As data confirm each link, the benign assumption gets weaker.  The figure is great and might be woven in sooner?

The discussion section might point out remaining gaps or reasons for lingering doubt that chronic suboptimal hydration increases chronic disease risk. The evidence base is not replete with randomized controlled trials in humans.  

In the abstract, wondered if the authors meant “engrained” instead of “engraved”.

In the abstract, consider softening to “mediate” instead of drives. Vasopressin pathways definitely ‘mediate’ disease mechanisms. ‘Driving’ seems a bit too strong, maybe attributing too much chronic disease effect to vasopressin, because vasopressin action is only one of many possible harmful effects of hyperosmotic stress.  A point that might get added as limitation to the discussion section.

Introduction

Line 38, might specify intake of drinking water or plain water intake

Lines 41-43, as currently worded, suggest that no-one has noticed the potential risk until now, which is not true. These sentences might be rephrased to draw attention to the debate or disagreement in the field over how to interpret compensation for mild dehydration or underhydration. People have reported on hyperosmotic stress as potential chronic disease risk factor, but the data have not been well-received;  There has been considerable resistance among physiologists, nephrologists and the food industry to the idea that elevated urine osmolality signals a problem (having experienced the flack from Reviewers on this point for years). Kenefick et al might be cited as expressing this point of view (Cheuvront, S.N.; Kenefick, R.W. Am I Drinking Enough? Yes, No, and Maybe. J. Am. Coll. Nutr. 2016, 35, 185–192). The compensation for mild dehydration (elevated urine osmolality) is considered a ‘normal’, healthy response to hyperosmotic exposure. Also, many researchers view body water retention that results from vasopressin system activity as desirable - they interpret the body water retention to be "better" hydration. An implication of thinking this way is that fewer people are detected as having sub-optimal hydration, leading to inconsistent prevalence estimates, and missed opportunities for intervention. 

Lines 46-47 skirt around the issue of no one definition of hydration. The paper might acknowledge that strong differences of opinion are related to the type of hydration disorder studied and different terminology OR this paper could just restrict the analysis to types of suboptimal hydration that trigger vasopressin (hyperosmotic dehydration or hyperosmolality with normal TBW)… Either way, something could be added about the particular focus to lines 46-49.

The section about Types of dehydration might explain how/why, for each type of hydration disorder,  vasopressin is triggered by cell shrinkage. 

Line 72. Kavouras did coin the term “underhydration” but Stookey defined the biomarker parameters cited, independently of Kavouras, in a paper that preceded ref 7 (doi: 10.3390/nu11030657).

Reference 11 reports a prevalence of elevated serum osmolality of 60% for the NHANES sample, so does ref 7. 

Lines 98-100 actually do not fit well in this paragraph about water intake relative to water intake recommendations, because these lines are about hydration biomarker data not water intake relative to cutoffs data. Maybe separate, gather ref 11 and ref 15 and mention as biomarker evidence that suggests that water intake may be inadequate. Reference 15 used a biomarker to classify hydration status (not a survey as suggested by line 107).

Section 4 could be a place to make a case for ‘habits’ by adding some literature that explain how the timing, type, volume of beverage intake is culturally/societally prescribed, institutionalized by things like cost, school meals and restaurant happy meal combos and portion size, adding some anthropology, economist, and psychology references.

Lines 155-168 imply that public health people have been aware of the beverage intake problem for enough years to get laws for taxation passed, i.e. people are already engaged in intervention efforts to change habits. The gap that might be highlighted in this paper (since people are already trying to intervene) is that intervention efforts may be slightly off-mark due to unappreciated importance of water.  In addition to efforts to decrease SSBs, we also need interventions to increase drinking water, in order to enjoy the health benefit of the intervention.

Suggesting to double check each use of “water” to see if clarification is needed regarding plain water or total water, and if the effect described is an effect of absolute increase in water or a relative increase (water went up, something else went down, total water stayed same).

Lines 260-261  made me wonder how is fructose absorbed into cells? In contrast to glucose uptake, which requires insulin. In healthy people with normal insulin response, glucose is not an effective osmolyte. Is fructose an “effective osmolyte”? are there some conditions when it is or is not an effective osmolyte?  (this is why mention of cell shrinkage is key earlier in the paper).

Line 174.. worth highlighting the knock out model data, somehow. Does this confirm the causal mechanism?

Arriving at the end of the paper, lines 359-367, I realized that the authors know what I mean about resistance to viewing the compensation as a problem. It would have helped me get into the paper sooner to know that from the beginning.

Thank you for this interesting paper.

Author Response

Reviewer 1

  1. This paper contributes to the hydration literature by helping to resolve controversy stemming from unchecked assumptions about homeostatic compensation for hyperosmolality. By succinctly summarizing data that show that the compensations are not benign, the paper warrants greater attention to indices of hyperosmotic stress and intervention potential to reduce chronic disease risk. Noting that the harms or costs related to compensation for hyperosmolality are preventable. A few edits might clarify the purpose and strengthen the contribution of this paper. The controversy and its implications (e.g. inconsistent findings, health professional inaction, missed intervention opportunities) might be introduced in the background section. The methods for or logic behind highlighting data points that demonstrate a harmful effect of compensation might be made more transparent. The introduction might outline or foreshadow what kinds of data will be presented or what data questions were considered to sway the reader’s opinion… Randomized trials, epidemiological data, plausible causal paths confirmed with knock out models. Low water or high salt à hyperosmolality-> vasopressin ->receptors-> pathways -> chronic disease Do we have evidence of each link in this causal chain? Is each of these steps in the mechanism prevalent in free-living populations?  If these are/were the authors implicit working questions, the paper might make that explicit. Explicit questions up front might give more context to how the headers fit together into the whole.  (Types of dehydration, what constitutes adequate water intake, causes of inadequate water consumption, intake of sweetened beverages, metabolic…effects of poor hydration) As data confirm each link, the benign assumption gets weaker.  The figure is great and might be woven in sooner? The discussion section might point out remaining gaps or reasons for lingering doubt that chronic suboptimal hydration increases chronic disease risk. The evidence base is not replete with randomized controlled trials in humans. 

 We thank the reviewer for her/his constructive comments and meticulous review. In this review, our objective was to bring interest in the “underhydration” phenomenon and argument the potential damage mediated by the chronic activation of two pathways that we have long studied, mostly from the basic research, vasopressin, and aldose reductase-fructose pathways. We recognize the topic is fascinating and deserves to be analyzed from different points, which we believe are beyond of the scope of the present review.

As per the reviewers’ advice, we have expanded the background.

  1. In the abstract, wondered if the authors meant “engrained” instead of “engraved”. In the abstract, consider softening to “mediate” instead of drives. Vasopressin pathways definitely ‘mediate’ disease mechanisms. ‘Driving’ seems a bit too strong, maybe attributing too much chronic disease effect to vasopressin, because vasopressin action is only one of many possible harmful effects of hyperosmotic stress. A point that might get added as limitation to the discussion section.

Thanks a lot, it is much better the word “ingrained” and we have changed it.  We also toned down and changed the word “drive” for “mediate”.

  1. Line 38, might specify intake of drinking water or plain water intake

We have clarified this point and specified “intake of drinking water”.

  1. Lines 41-43, as currently worded, suggest that no-one has noticed the potential risk until now, which is not true. These sentences might be rephrased to draw attention to the debate or disagreement in the field over how to interpret compensation for mild dehydration or underhydration. People have reported on hyperosmotic stress as potential chronic disease risk factor, but the data have not been well-received; There has been considerable resistance among physiologists, nephrologists and the food industry to the idea that elevated urine osmolality signals a problem (having experienced the flack from Reviewers on this point for years). Kenefick et al might be cited as expressing this point of view (Cheuvront, S.N.; Kenefick, R.W. Am I Drinking Enough? Yes, No, and Maybe. J. Am. Coll. Nutr. 2016, 35, 185–192). The compensation for mild dehydration (elevated urine osmolality) is considered a ‘normal’, healthy response to hyperosmotic exposure. Also, many researchers view body water retention that results from vasopressin system activity as desirable - they interpret the body water retention to be "better" hydration. An implication of thinking this way is that fewer people are detected as having sub-optimal hydration, leading to inconsistent prevalence estimates, and missed opportunities for intervention.

Thanks a lot for taking notice of this omission. After the marked sentence, we have included two references.

  1. Lines 46-47 skirt around the issue of no one definition of hydration. The paper might acknowledge that strong differences of opinion are related to the type of hydration disorder studied and different terminology OR this paper could just restrict the analysis to types of suboptimal hydration that trigger vasopressin (hyperosmotic dehydration or hyperosmolality with normal TBW)… Either way, something could be added about the particular focus to lines 46-49.

Thanks, we have focused on the topic of the review.

  1. The section about Types of dehydration might explain how/why, for each type of hydration disorder, vasopressin is triggered by cell shrinkage.

Thanks for the comment. We believe that the physiology of vasopressin action has already been extensively covered. In the present review besides the vasopressin system participation, we wanted to point out the participation of aldose reductase-fructose pathway and the interaction and synergy between the two systems, as those mechanisms are more recently described and implicated with the various alterations induced by its chronic activation.

  1. Line 72. Kavouras did coin the term “underhydration” but Stookey defined the biomarker parameters cited, independently of Kavouras, in a paper that preceded ref 7 (doi: 10.3390/nu11030657).

We are very grateful for this important precision. We have now corrected this and recognized Stookey as the author who defined the biomarker parameters to consider an “underhydration” state.

  1. Reference 11 reports a prevalence of elevated serum osmolality of 60% for the NHANES sample, so does ref 7Moreover. Lines 98-100 actually do not fit well in this paragraph about water intake relative to water intake recommendations, because these lines are about hydration biomarker data not water intake relative to cutoffs data. Maybe separate, gather ref 11 and ref 15 and mention as biomarker evidence that suggests that water intake may be inadequate. Reference 15 used a biomarker to classify hydration status (not a survey as suggested by line 107).

Thanks for your advice. We have reworked this paragraph accordingly to your suggestions.

  1. Section 4 could be a place to make a case for ‘habits’ by adding some literature that explain how the timing, type, volume of beverage intake is culturally/societally prescribed, institutionalized by things like cost, school meals and restaurant happy meal combos and portion size, adding some anthropology, economist, and psychology references.

As per Academic Editor´s advice, we have merged this section with section 2.

  1. Lines 155-168 imply that public health people have been aware of the beverage intake problem for enough years to get laws for taxation passed, i.e. people are already engaged in intervention efforts to change habits. The gap that might be highlighted in this paper (since people are already trying to intervene) is that intervention efforts may be slightly off-mark due to unappreciated importance of water. In addition to efforts to decrease SSBs, we also need interventions to increase drinking water, in order to enjoy the health benefit of the intervention.

We have already included a comment regarding recommendations to encourage a greater intake of plain water in the Conclusions section.

 Suggesting to double check each use of “water” to see if clarification is needed regarding plain water or total water, and if the effect described is an effect of absolute increase in water or a relative increase (water went up, something else went down, total water stayed same).

Thanks, done

  1. Lines 260-261 made me wonder how is fructose absorbed into cells? In contrast to glucose uptake, which requires insulin. In healthy people with normal insulin response, glucose is not an effective osmolyte. Is fructose an “effective osmolyte”? are there some conditions when it is or is not an effective osmolyte? (this is why mention of cell shrinkage is key earlier in the paper).

Fructose is absorbed via transporters into the cells (GLUT5 or GLUT2) and does not require insulin. Studies in humans demonstrated that fructose, contrary to glucose, can induce vasopressin secretion by activating a short-term osmotic mechanism and a second stimulation is produced through non-osmotic mechanisms (1). Our studies have found that fructose metabolism in hypothalamic neurons via fructokinase is a fundamental step for vasopressin secretion as fructokinase knock out mice fail to secrete vasopressin when acutely dehydrated. We have already described this mechanism in the review.

  1. Line 174.. worth highlighting the knock out model data, somehow. Does this confirm the causal mechanism?

Thanks. We believe that the basic science data is adequately highlighted for the purposes of this review.

  1. Arriving at the end of the paper, lines 359-367, I realized that the authors know what I mean about resistance to viewing the compensation as a problem. It would have helped me get into the paper sooner to know that from the beginning. Thank you for this interesting paper.

We reiterate our appreciation for a thoughtful revision of our manuscript and excellent comments and suggestions.

Reviewer 2 Report

  1. There are a lot of double spaces all along with the text. Please correct it.
  2. The introduction section should be more explained and developed.
  3. The first paragraph of the “type of dehydration” section, misses references all along with the sentences. Please correct it.
  4. Line 71, misses a final point after 135 mEq/L.
  5. Figure 1: Do the researchers have a license for using BioRender images for scientific publication purposes? This is a very important point that should be guaranteed.

Author Response

Reviewer 2

Comments and Suggestions for Authors

1. There are a lot of double spaces all along with the text. Please correct it.

Thanks. Done.

2. The introduction section should be more explained and developed.

As per reviewer´s suggestions, we have expanded the introduction

3. The first paragraph of the “type of dehydration” section, misses references all along with the sentences. Please correct it.

Thanks a lot for this observation. We have corrected the paragraph and included references.

4. Line 71, misses a final point after 135 mEq/L.

Thanks. Done.

5. Figure 1: Do the researchers have a license for using BioRender images for scientific publication purposes? This is a very important point that should be guaranteed.

Thanks. We have a license for using Biorender.